# The Role of Ultrasound as a Diagnostic and Therapeutic Tool in Experimental Animal Models of Stroke: A Review

**DOI:** 10.3390/biomedicines9111609

**Published:** 2021-11-03

**Authors:** Mari Carmen Gómez-de Frutos, Fernando Laso-García, Iván García-Suárez, Luke Diekhorst, Laura Otero-Ortega, María Alonso de Leciñana, Blanca Fuentes, Dolores Piniella, Gerardo Ruiz-Ares, Exuperio Díez-Tejedor, María Gutiérrez-Fernández

**Affiliations:** 1Neurological Sciences and Cerebrovascular Research Laboratory, Department of Neurology and Stroke Centre, La Paz University Hospital, Neuroscience Area of IdiPAZ Health Research Institute, Universidad Autónoma de Madrid, 28046 Madrid, Spain; mcarmen.gomezf@gmail.com (M.C.G.-d.F.); fernilaso.9@gmail.com (F.L.-G.); artedo05@yahoo.es (I.G.-S.); luke.diekhorst@gmail.com (L.D.); oteroortega.l@gmail.com (L.O.-O.); malecinanacases@salud.madrid.org (M.A.d.L.); blanca.fuentes@salud.madrid.org (B.F.); dolores.piniella.alcalde@idipaz.es (D.P.); adalbertog.ruiz@salud.madrid.org (G.R.-A.); 2PhD Program in Neuroscience, Department of Anatomy, Histology and Neuroscience, Universidad Autónoma de Madrid-Cajal Institute, 28029 Madrid, Spain; 3Emergency Service, San Agustín University Hospital, 33401 Avilés, Spain

**Keywords:** animal model, hemorrhagic, ischemic, stroke, ultrasound

## Abstract

Ultrasound is a noninvasive technique that provides real-time imaging with excellent resolution, and several studies demonstrated the potential of ultrasound in acute ischemic stroke monitoring. However, only a few studies were performed using animal models, of which many showed ultrasound to be a safe and effective tool also in therapeutic applications. The full potential of ultrasound application in experimental stroke is yet to be explored to further determine the limitations of this technique and to ensure the accuracy of translational research. This review covers the current status of ultrasound applied to monitoring and treatment in experimental animal models of stroke and examines the safety, limitations, and future perspectives.

## 1. Introduction

Ultrasound has several well-known advantages, including its noninvasiveness, low cost, portability, safety, and rapid real-time imaging [1,2]. Point-of-care ultrasound therefore became an important modality for clinically evaluating critically ill and injured patients in emergency medicine [3,4,5]. Ultrasound could be an essential tool in many neurological diseases, particularly in stroke, in which “time is brain” and “imaging is brain” [6]. Ultrasound imaging not only enables the study of anatomical structures but also provides useful information on the organs’ function, blood vessel characteristics, and blood flow (Doppler imaging) [7]. Ultrasound can therefore help speed the decision-making process and improve and accelerate the diagnosis and treatment, resulting in reduced neurological sequelae.

Due to the enormous potential of ultrasound, we reviewed its application in experimental animal models of stroke (ischemic stroke and intracerebral hemorrhage [ICH]), in monitoring and therapeutics (Figure 1), and assessed its safety, limitations, and future perspectives.

For this review, we conducted a search on the Google Scholar, PubMed, and grey literature databases, employing the search terms ultrasound, ultrasonography, Doppler, microbubbles, stroke, ischemic stroke, hemorrhagic stroke, rat, animals, animal model, treatment, therapeutic, diagnostic, safety, limitations, brain, ultrasound frequency, low-frequency ultrasound, and high-frequency ultrasound. We restricted the search to animal studies, ultrasound, and stroke; however, no restriction was applied to the year of publication.

## 2. Ultrasound Applications in Ischemic Stroke

Ischemic stroke is the most common stroke type [8] and it occurs when a blood vessel is occluded by a thrombus or embolism, leading to loss of oxygen and glucose in cerebral tissue [9]. The resulting cerebral ischemia causes neuronal damage in the core zone shortly after onset. Surrounding the core zone is an area known as the penumbra [10], where the cells are not yet irreversibly damaged and can be recovered if cerebral blood flow (CBF) is restored [9]. Ultrasound can therefore play an important role in monitoring ischemic stroke.

### 2.1. Ultrasound Monitoring in Ischemic Stroke

In a 1999 study conducted on focal cerebral ischemia in rabbits, Els T et al. demonstrated the usefulness of ultrasound for the continuous monitoring of CBF velocity to follow rapid changes in the dynamics of the ischemic process [11], thereby offering a tool for stroke control from early monitoring to therapy. These changes in CBF were studied using contrast-enhanced ultrasound (CEUS) [12], Doppler velocimetry [13], and functional ultrasound [14]. Ultrasound contrast agents are encapsulated microbubbles in suspension with great echogenicity that can enhance the echo signals [15,16]). These microbubbles accumulate in the microvasculature providing anatomical reference information or molecular characteristics at target sites [15,16]. Microbubbles can be filled with air or gas. The most common gases are sulfur hexafluoride, decafluorobutane, perfluorobutane, perflutren, and octafluoropropane [15,16]. Microbubbles also need a cover to reduce gas diffusion into the blood. This cover can be made from polymers, lipids, or denatured albumin [15]. Along with the microbubble cover, other compounds like PEG (polyethylene glycol) can be used to enhance stability and reduce immune system recognition. Despite microbubble are the most frequently used, other options, such as perfluorocarbon nanoparticles or echogenic liposomes, are being studied to improve ultrasound imaging [15]. Premilovac D et al. applied transcranial CEUS to rats with middle cerebral artery occlusion (MCAO) to measure cerebral perfusion after stroke in real-time. In this noninvasive manner, the authors observed a reduction in cerebral perfusion during ischemia and hyperperfusion during recanalization [12].

Doppler ultrasound made it possible to determine blood flow velocities in rats with ischemic stroke [17,18]. Specifically, ultrasound imaging of the circle of Willis made it possible to obtain information on cerebral collateral recruitment and thus the presence or absence of ischemic lesions, reflow, hemodynamic changes, and reperfusion, all of which can clarify the underlying recovery mechanism after stroke [17,18]. Functional ultrasound is a useful imaging technique for studying brain perfusion during and after MCAO in rats with acute and postacute stroke [14,19]. In a thromboembolic MCAO model of mice treated with tissue plasminogen activator (tPA), the ultrafast ultrasound study of brain reperfusion in the ischemic lesion helped predict the prognostic outcomes and thrombolytic treatment response [20]. There is also an enhanced form of Doppler ultrasound known as the triplex ultrasonographic mode, which can be useful for studying blood flow in experimental animal models of stroke [13] (Table 1). This evidence indicates that ultrasound is a feasible, effective, and rapid method for evaluating/monitoring the dynamics of the ischemic process in experimental studies.

### 2.2. Therapeutic Ultrasound in Ischemic Stroke

In a 1974 study by Sobbe et al., ultrasound was employed to recanalize thrombosed femoral arteries in dogs, obtaining good results with no complications [40]. The application of low- or high-intensity pulsed ultrasound in an MCAO rat model opens possibilities for developing new therapies [21,22]. Specifically, low-intensity pulsed ultrasound (0.5 MHz) might have protective effects and promote CBF [21], while 1 MHz can trigger the expression of brain-derived neurotrophic factor (BDNF), reduce brain damage [23], and reduce neuronal apoptosis [24]. Low-intensity pulsed ultrasound (2 MHz) can reduce brain oedema and ischemic damage [24], thus facilitating the recovery of ischemic areas. In a mouse stroke model after the administration of human bone marrow mesenchymal stem cells, ultrasound was shown to promote neuronal differentiation, improving functional recovery, and reducing infarct volume [25]. 

Ultrasound is known to enhance thrombolytic drug activity [41,42] and to improve clot lysis. High-intensity focused ultrasound (1.5 MHz) initiates thrombolysis safely and effectively in rabbits subjected to an embolic stroke [43]. Low-intensity ultrasound (25.570 Hz) was employed in parallel to tPA infusion in a rat model of embolic ischemic stroke. The combination was more effective than tPA treatment alone in reducing infarct volume [26]. The administration of microbubbles, with or without tPA, can further enhance ultrasound thrombolysis [27,44,45]. Gao S et al. reported that intravenous microbubble administration (without tPA) and the use of ultrasound improved CBF by recanalizing the ipsilateral carotid artery of pigs [28]. This strategy also reduced infarct volume in rabbits with internal carotid embolization [29] and reduced glutamate concentration, suggesting a protective role for ultrasound and microbubbles in rats with acute ischemic stroke [30]. The combination of tPA and microbubble+ultrasound in a rabbit arterial embolic stroke model led to a reduction in infarct volume and hemorrhage formation [27] and improved microvascular patency in rats after MCAO [31]. Culp WC et al. demonstrated that the intravenous administration of microbubbles loaded with a glycoprotein 2b/3a receptor antagonist successfully augmented the effect of transcutaneous low-frequency ultrasound (1 MHz) in dissolving acute intracranial thrombi without thrombolytic drugs in a swine stroke model [32]. 

Ultrasound-targeted microbubble destruction (UTMD) is a technique that uses low-frequency moderate-power ultrasound in combination with microbubbles that encapsulate drugs to enhance drug delivery into tissues. Ultrasound can increase microvessel permeability and trigger cavitation, thus allowing transport of microbubbles and release of drugs into the desired area [46,47]. 

Moreover, one of the applications of ultrasound in the brain is the temporary opening of the blood brain barrier (BBB) [48]. The detailed mechanism behind ultrasound-mediated BBB disruption is still under investigation, but two major mechanisms were proposed as an explanation. Both stable and inertial cavitation together with microbubbles can help open the BBB. However, whereas stable cavitation disrupts the BBB via wobbled expansion and contraction of microbubbles, inertial cavitation is caused by the violent collapse of the microbubbles, which could lead to vascular damage [49]. The first study of ultrasound-induced BBB opening showed small lesions with severe parenchymal damage [50]. Recent research has indicated that low-frequency and focused ultrasound noninvasively opens the BBB and increase expression of caveolin, which plays an important role in the transport of large vesicles [51,52]. The duration of BBB disruption is also under investigation and might be related to the diameter of the delivered substances. Small particles of 1 nm can cross several hours after BBB disruption, with complete BBB restoration after 24 h; 4 nm particles stopped crossing the BBB after 2 h [53]. Regardless of the duration of the BBB opening, cavitation allows large molecules and genes to cross the plasma membranes of cultured cells after the application of acoustic energy [54,55]. These findings improve our understanding of how ultrasound influences the BBB and help the development of new drugs. 

Drug-loaded microbubbles can be excited and broken by ultrasound thus releasing substances to desired targets and, increasing the local drug concentration. For example, microbubble encapsulation and ultrasound delivery of BDNF to the lesion site improves functional recovery, which is associated with tract connectivity restoration, remyelination, and brain recovery markers in a rat model of ischemic stroke [33]. The release of phosphatidylserine contained in microbubbles using ultrasound activated microglia during neuroinflammation after stroke in rats [34]. UTMD could be also used in gene therapy. For instance, it was applied in mice after transient MCAO for delivery of *VEGF* (Vascular Endothelial Growth Factor) gene and demonstrated a reduction in ischemic injury [35] (Table 1). 

## 3. Ultrasound Applications in Intracerebral Hemorrhage

ICH is produced by blood extravasation into the brain [39], with a significant increase in volume growth after the first few hours [36,56]. ICH lacks effective treatments, leading to high mortality and morbidity [36,39] and producing severe neurological outcomes. Therefore, the early detection of ICH and the development of effective therapeutic approaches assume great importance, with ultrasound-based technologies opening up a promising field of research and treatment.

### 3.1. Ultrasound Monitoring in Intracerebral Hemorrhage

Few studies employed it in an experimental animal model of ICH. Doppler can be used in ICH to study tissue perfusion in animal models [37]. To our knowledge, the study by Ke Z et al. [37] is the only one to use high-frequency Doppler ultrasound (4–13 MHz) to analyze blood flow velocity changes in certain arteries in rats before and after ICH. As the authors noted in the study, this tool could be employed in future studies combined with protective agents that affect cerebral hemodynamics after ICH [37] (Table 1).

Despite the use of ultrasound in ischemic stroke, ultrasound is underutilized in the monitoring of ICH, resulting from a lack of scientific evidence in this field. B-mode ultrasound could be used to monitor hematoma volume in experimental models of ICH. 

### 3.2. Therapeutic Ultrasound in Intracerebral Hemorrhage

Only a few studies employed ultrasound as treatment for ICH. Repeated applications of ultrasound in rats following ICH were shown to promote brain angiogenesis by enhancing the expression of extracellular matrix molecules (such as collagen and integrins) and accelerating neurological recovery [36]. Stroick M et al. demonstrated that ultrasound combined with microbubbles did not increase the hematoma volume, brain oedema extent, or apoptosis rate in rat brains during ICH [38]. Moreover, in a study conducted in dogs with ICH, the tandem ultrasound-microbubbles was used visualizing successfully the hematoma through the bone window [39] (Table 1).

## 4. Safety of Ultrasound

Safety is the primary consideration when applying a technique in *in vivo* research. Brain barriers are crucial in neurological diseases. Preclinical studies in animals provide an overwhelming advantage when evaluating the safety of ultrasound exposure, given that there are various methods to evaluate the BBB integrity that cannot be performed on humans. The most commonly used technique to assess the opening of the BBB in preclinical studies was the administration of Evans Blue [57].

This chemical is intravenously administered during exposure with ultrasound and its extravasation to the brain parenchyma is studied using a fluorescence microscope postmortem [33]. More advanced techniques allow for the observation of BBB integrity *in vivo*. The use of a gadolinium-based contrast can be imaged by magnetic resonance imaging (MRI), assessing BBB opening after ultrasound exposure and evaluating the extravasation of this agent into the brain parenchyma [58]. Lastly, hemorrhagic transformation is an important concern to consider because ultrasound exposure could disrupt the integrity of vessels in the brain. The presence of hemorrhage can be analyzed in animal models by using hematoxylin and eosin, Nissl staining in postmortem tissue and by MRI in vivo [59]. Immunohistochemistry and immunofluorescence are techniques used to evaluate the safety of ultrasound exposure by analyzing the integrity of the blood vessels by using antibodies against ZO-1 (Zona Occludens-1) or lectin markers [33]. These techniques also allow us to analyze apoptosis, necrosis, abnormal cell migration and membrane dysfunction in the parenchyma using a fluorescence microscope postmortem. 

Several parameters are important to safety, such as frequency, intensity, mechanical index, duty cycle or pulse repetition frequency, no consensus was reached. Since different settings are used for monitoring and treatment using ultrasound, the safety concerns will be treated separately. 

### 4.1. Safety Data of Ultrasound in Monitoring Stroke

Several studies using ultrasound imaging in animal models have reported no tissue damage using a 6.3 MHz linear transducer [17], 12 MHz linear transducer [18] and 15 MHz linear array transducer [14] in MCAO models. Other studies applied imaging ultrasound in an embolic stroke model using a 12 MHz linear transducer and did not report secondary effects [60]. Moreover, imaging ultrasound was also applied in an animal model of ICH using 13–14 MHz transcranial Doppler ultrasonography and did not report secondary hemorrhages [37]. Other studies examined the brain’s blood flow and anatomical structures in healthy animals using a 13 MHz linear transcranial transducer [61] and 16 MHz transducer [62]. Considering these studies, the range of 12–16 MHz frequencies to be safe for imaging in the animal brain. However, other frequencies of ultrasound, such as 20 MHz, were shown to initiate cellular injury by inactivating several enzymes and causing free radical production in the brain [63], and ultrasound exposure of 3 MHz increased antioxidant enzyme activities in the rat brain [64]. Therefore, demonstrating safety at the 12–16 MHz frequency range does not directly lead to safety at lower frequencies. Further studies investigating the safety of ultrasound during imaging would help set limits on the machine output parameters and determine which frequency range is adequate and safe to use.

### 4.2. Safety Data of Therapeutic Ultrasound in Stroke

When ultrasound is used as a treatment, the waves need more energy compared with that of imaging applications. The absorption of energy within tissue leads to a rise in temperature. This phenomenon is known as the thermal effect [65]. The magnitude and duration of the temperature increase may cause tissue damage. Ultrasound waves also lead to mechanical effects that result in pressure changes, shear forces, and cellular damage with molecule release [65]. 

Lower frequencies demonstrated safety in experimental animal models of stroke: 0.5 MHz may have protective effects and promote CBF [21], 1 MHz can reduce brain damage [23] and neuronal apoptosis [24], and 2 MHz can reduce brain oedema and ischemic injury [22]. For ultrasound enhanced thrombolysis frequencies of 1.5 MHz [43] was shown to be safe. 

Some studies analyzed how microbubble cavitation threshold can be reduced when combined with ultrasound [66]. Cavitation is indicated by the mechanical index [67], this is another important parameter to take into account for the safety of the technique. Although the optimal mechanical index to be used is yet to be determined [68], the British Medical Ultrasound Society established a mechanical index limit of 0.7 for microbubble cavitation [69]. Not only is the mechanical index important when evaluating the safety of ultrasound-induced microbubble cavitation as a treatment for stroke, but also the frequency is a relevant parameter. The first study that demonstrated the safety of combined ultrasound and microbubbles after acute cerebral ischemia in rats was performed by Fatar M et al. in 2008 and used 2 MHz of frequency [30]. Since then, ultrasound cavitation of microbubbles was also shown to not induce secondary hemorrhage in animal models of ischemia using frequencies between 1.6 and 7 MHz [53,55,70], and in models of hemorrhage using 2 MHz [38]. 

Another relevant safety aspect to consider is the intensity. Intensities 57 and 86 mW/cm^2^ [21], 128 mW/cm^2^ [22], and 528 mW/cm^2^ [23] used as a therapy to promote brain protection and reduce ischemic damage were found to be safe in an experimental animal model of stroke. 

The use of continuous Doppler mode was shown to induce greater tissue damage than the pulsed application in animal models. Schneider F et al. showed that transcranial application of 20 kHz continuous ultrasound as a treatment in rats caused vasogenic and cytotoxic brain oedema and a loss of neuronal cells, resulting in a higher death cell rate [71]. To prevent brain damage, as mentioned before, pulsed mode emerged as an option to reduce the adverse effects in terms of mortality, lesion volume and recovery [72]. The pulse repetition frequency of each mode should also be taken into consideration. In one study, the ischemic core was exposed to an ultrasound signal for 360 trials of 0.4 s every 10 s, which did not produce any adverse effects [21]. Another study targeted ultrasound on the injured cortical areas in rats every 5 min for 15 min per day for a maximum of 5 days, which also did not produce any adverse effects [23]. Again, no adverse effects were found when rats received 1 ms ultrasound bursts every second for 20 s [43]. Another study directed ultrasound waves at the left hemisphere for 5 min, giving ultrasound bursts every 500 ms; no adverse effects were detected [35]. No tissue damage was detected when using ultrasound with a duty cycle of 20% for 1 h in parallel to recombinant tPA infusion [26]. 

## 5. Limitations and Future of Ultrasound

### 5.1. Limitations of Ultrasound

Despite all its advantages, ultrasound currently has certain limitations. The technique requires the knowledge and expertise of a well-trained operator [2], and therefore caution is necessary when acquiring and interpreting images. A standardization of ultrasound parameters is also needed, given that several studies have failed to report the intensity, duration and mechanical index employed, hindering the studies’ reproducibility [73]. Another limitation of ultrasound is its reduced penetration in the presence of bone structures [2,74]. In certain studies, this limitation was countered by thinning or removing the skull [14,75], which causes additional damage. The lack of scientific articles on the use of ultrasound in experimental models of stroke also highlights the need for further research in this area, especially in ICH.

### 5.2. Future of Ultrasound in Preclinical Models

#### 5.2.1. New Approaches of Ultrasound

The information on brain anatomy provided by ultrasound may help research on neurological diseases. B-mode echography was proven as a feasible tool for noninvasively studying anatomical structures of the rat brain [61] and, Superb Microvascular Imaging (SMI), an advanced and innovative ultrasound mode based on Doppler, allows study of microvascular blood flow with and without contrast agents, enabling the creation of an innovative vascular brain map [61]. 

Technological advances such as “ultrafast” ultrasound systems can process thousands of frames per second. The study conducted by Macé E et al. employed this technique to quantify blood perfusion differences in the rat brain [76]. Ultrasound localization microscopy (a super-resolution ultrasound imaging based on ultrafast ultrasound) increased sensitivity and image definition, enabling the study of cerebral blood volumes in mice after stroke [20]. This ultrasound modality could become an interesting tool for obtaining deep super-resolution vascular images and providing improved transcranial imaging. 

Tan JK et al. demonstrated that gene transfer after prior administration in the intraventricular area could be increased by the use of ultrasound through a temporal opening of the choroid plexus epithelium [77]. The combined use of ultrasound-mediated BBB opening and microbubbles for drugs or gene delivery could be further developed for future clinical applications. Currently, several clinical trials are underway assessing the safety of BBB disruption in a variety of diseases [78]. Lastly, stem cell-based therapies offer a promising view of stroke treatment. In this field, a new therapeutic approach for ultrasound could be based on the biological effect of focus ultrasound in increasing the efficacy of stem cell homing in the brain parenchyma after their intravenous administration [79]. 

These advances could enhance the widespread use of ultrasound and its variants in future preclinical studies, as well as in the clinic. The versatility of ultrasound and its prospects provide the option of obtaining data in real-time and could lead to shorter delays to treatments for various neurological diseases.

#### 5.2.2. Imaging 3D/4D in Ultrasound 

Other novel techniques, such as 3D and 4D model ultrasound, could emerge as powerful modalities of choice in the diagnosis of various diseases [80]. Work performed on rodents [81,82], pigeons [83], and ferrets [84] were able to construct 3D maps of brain activity and to obtain 3D vascular imaging. Additionally, we also find ultrasound neuroimaging in 4D. In preclinical studies, 4D ultrasound microvascular imaging could become a valuable tool for studying brain hemodynamics, cerebral flow autoregulation and vascular remodeling after stroke [85]. Also, it was demonstrated that 4D functional ultrasound can be used to obtain 4D functional connectivity in rats [86]. These results suggest a possible new modality for neuroimaging of the brain that could expand the observation and study of the whole brain in animal models

## 6. From Preclinical Models to Clinical Practice 

In humans, ultrasound is used extensively and the possibility of developing new advances for monitoring or therapeutic purposes in animals may help to expand its use in the clinic. Being able to use ultrasound in the laboratory will accentuate the similarity between the two scenarios, which is fundamental in translational research.

Ultrasound is widely used as a diagnostic tool in patients with ischemic stroke [7]. However, is not routinely employed in patients with ICH, despite its potential role in the measurement of ICH volume and growth [1,39,87,88].

In clinical practice, ultrasound was used ad and adjuvant to thrombolysis, the so-called sonothrombolysis. This approach improves arterial recanalization and functional outcomes. [89]. However, there is a shortfall in the translation of therapeutic studies with microbubbles. Currently, in vitro and in vivo studies are exploring the possibility to enhance transgene expression and targeted drug delivery without adverse effects [90]. For this reason, development in ultrasound application with microbubbles will allow new therapeutical approaches for stroke patients.

## 7. Conclusions

Ultrasound is a noninvasive, low-cost, rapid, and safe technique whose versatility makes it an imaging tool worth considering for stroke, given that stroke requires rapid and appropriate action. In preclinical studies of stroke (ischemic stroke and ICH), ultrasound proved to be useful in monitoring and as a therapeutic tool, and its safety profile was demonstrated in various experimental animal models. At present, however, only a few studies were performed on its use in stroke, particularly regarding ICH. The lack of scientific data on the application of ultrasound in the preclinical field of stroke necessitates continuing the study of this promising imaging technique to solve its present limitations and to safely transfer the results from bench to bedside.

## Figures and Tables

**Figure 1 biomedicines-09-01609-f001:**
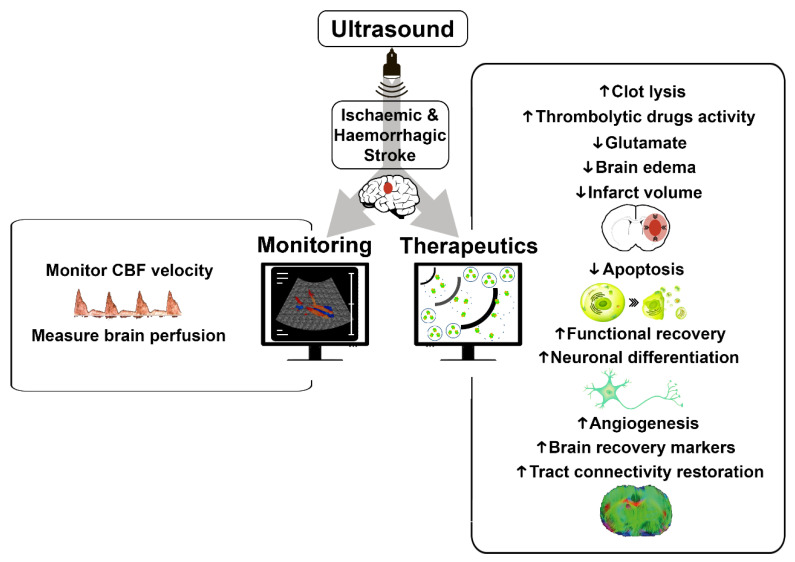
Potential of ultrasound in animal models of stroke. Ultrasound applied to brain after stroke can provide rapid and additional information for monitoring or as a specific and local treatment. Main effects of ultrasound in monitoring and as a therapeutic tool are summarized in the figure. CBF: cerebral blood flow. Up arrows mean increase and down arrows mean decrease.

**Table 1 biomedicines-09-01609-t001:** Main experimental animal models of stroke (ischemic stroke and intracerebral hemorrhage) using ultrasound with monitoring and therapeutic application.

ISCHEMIC STROKE							
Reference	Sex/Species/n	Stroke type	Ultrasound	Frequency	Duration	Application	Findings
Els T. [11]	M/New Zealand rabbits/9	MCAO	Transcranial Doppler	4 MHz	Burst interval of 2 s	Monitoring: CBF velocity measurements	Decrease in CBF in the affected hemisphere
Premilova D. [12]	M/SD rats/16	MCAO	Transcranial CEU	15.2 Hz	3 times, 60 s	Monitoring: CBF velocity measurements	Hyperemia poststroke
Brunner C. [14]	M/SD rats18	MCAO	fUS	15 MHz	3 min	Monitoring: CBF velocity measurements	Mapping changes of CBF in stroke
Li L. [17]	M/SD rats/12	MCAO	Transcranial Doppler ultrasonography	6.3 MHz	--	Monitoring: CBF velocity measurements	Redistribution of CBF after stroke
Bonnin P. [18]	M&F/Wistar rats/92	MCAO	Colour Doppler	12 MHz	15 min	Monitoring: CBF velocity measurements	Collateral CBF protective role
Brunner C. [19]	M/SD rats/12	MCAO	fUS	1.43 MHz	30 min (baseline), 90 min (occlusion period) and 90 min (after clip removal)	Monitoring: CBF velocity measurements	Contralesional CBF response after stroke
Hingot V. [20]	M/Swiss mice/19	MCAO	Ultrafast	500 Hz	--	Monitoring: CBF measurements	Hypoperfusion in ischemic lesion after stroke
Guo T. [21]	M/SD rats/38	MCAO	pTUS	0.5 MHz	60 min	Therapeutic: protective role of ultrasound	Promotion of CBF, decrease of ischemic lesion
Alexandrov AV. [22]	--/Long-Evans rats/32	MCAO	Pulsed-wave transcranial	2 MHz	90 min	Therapeutic: effect on infarct volume	Reduction of ischemic brain damage and oedema Promotion of microcirculation
Chen CM. [23]	M/C57BL/6 mice/18	MCAO	Low-intensity pulsed	1 MHz	15 min daily	Therapeutic: protective effect	Reduction of brain damage VEGF levels recovered Increase BDNF protein expression
WU CT. [24]	M/C57BL/6 mice/40	MCAO	Low-intensity pulsed	1 MHz	15 min daily	Therapeutic: prevention of recurrent stroke	Reduction of lethality rate Prevent histopathological changes in brain
Cho SE. [25]	M/C57BL/6N mice/30	Photothrombosis	--	0.04 MHz	20 min/day	Therapeutic: evaluation of ultrasound effect in vitro and *in vivo*	Promotion of neuronal differentiation and neurogenesis
Daffertshofer M. [26]	--/Wistar & SD rats/47	Embolic stroke	Low-intensity	22.570 Hz	1 h	Therapeutic: ultrasound with tPA	Reduction of infarct volume
Brown AT. [27]	--/New Zealand rabbits/--	Emboli	--	1 MHz	60 min	Therapeutic: ultrasound and MB with tPA	Significant clot lysis. Reduction of infarct volume and ICH
Gao S. [28]	--/Pigs/--	Bilateral ICAO	Transcranial	1.6 MHz	5 and 20 µs	Therapeutic: ultrasound w/o MB	Improvement in CBF
Culp WC. [29]	--/New Zealand Rabbits/74	Internal carotid embolization	Pulsed-wave	1 MHz	1 h	Therapeutic: ultrasound and MB or tPA	Reduction of infarct volume
Fatar M. [30]	M/Wistar rats/16	MCAO	TCCD	2 MHz	30 min	Therapeutic: ultrasound and MB	Reduction of infarct volume
Schleicher N. [31]	M/Wistar rats/36	MCAO	Transcranial	3 MHz	60 min	Therapeutic: ultrasound and MB with tPA	Improvement of microvascular patency
Culp WC. [32]	--/Pigs/15	Autogenous thrombus	LFUS	1 MHz	24 min	Therapeutic: ultrasound and MB with glycoprotein	Augmented thrombolysis
Rodríguez-Frutos B. [33]	M/SD rats/143	Subcortical stroke	UTMD	7 MHz	5 min	Therapeutic: ultrasound and MB with trophic factor	Functional recovery. Fiber connectivity restoration Increased brain marker expression
Zhao R. [34]	M/SD rats/30	Intraluminal MCA blockage	UTMD	1.03 MHz	60 s	Therapeutic: ultrasound and MB with phosphatidylserine	Reduction in time of BBB opening Produced an early activated microglia/macrophage
Wang HB. [35]	F/CD1 mice/--	Transient MCAO	UTMD	1.6 MHz	5 min	Therapeutic: ultrasound and MB with VEGF gene	Infarct areas and apoptosis reduction Increased vessel density
**HEMORRHAGIC STROKE**							
**Reference**	**Species**	**Stroke type**	**Ultrasound type**	**Frequency**	**Duration**	**Application**	**Findings**
Mu HM. [36]	M/SD rats/280	ICH	Therapeutic	1 MHz	15 min, once daily	Therapeutic: effects on brain angiogenesis	Increase expression of integrins and collagen Microvessels formation promotion Functional recovery
Ke Z. [37]	M/SD rats/18	ICH	Transcranial Doppler	4–13 MHz	--	Monitoring: CBF velocity measurements	Flow velocity changes after stroke
Stroick M. [38]	M/Wistar rats/14	ICH	Transcranial	2 MHz	30 min	Therapeutic: ultrasound and MB safety	Ultrasound + MB due not increase additional damage
Zhou X. [39]	M/Dogs/12	ICH	Contrast-enhanced ultrasound	7 MHz	Every 30 min until hematoma formed	Monitoring: hematoma visualization	Brain hemorrhage imaging

**Abbreviations:** BBB, blood brain barrier; CBF, cerebral blood flow; CEU, contrast-enhanced ultrasound; F, female; fUS, functional ultrasound; ICAO, internal carotid artery occlusion; ICH, intracerebral haemorrhage; LFUS, low-frequency ultrasound; M, male; MB, microbubbles; MCA, middle cerebral artery; MCAO, middle cerebral artery occlusion; MHz, megahertz; pTUS, pulsed transcranial ultrasound; SD, Sprague-Dawley; TCCD, transcranial color-coded duplex sonography; tPA, tissue plasminogen activator; UTMD, ultrasound-targeted microbubble destruction; --, not indicated.

## Data Availability

Not applicable.

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
