# Peer review of "The Role of Ultrasound as a Diagnostic and Therapeutic Tool in Experimental Animal Models of Stroke: A Review"

_biomedicines, 2021, doi:10.3390/biomedicines9111609_

Round 1

Reviewer 1 Report

This is a very interesting topic. Overall the review is well written and provides a rather comprehensive review on the potential use of ultrasound in animal models of stroke. 

Having said that, there is a MAJOR issue in the way the paper is written, in multiple places in the manuscript. The use of ultrasounds is very useful (an this is the topic of the review as reflected in the title), in animal models of stroke, including for testing drugs when the BBB has been opened by ultrasounds. However, when translating to humans, there are many major limitations, including the bones (the authors even clearly identify this in rodents with the need of thinning the bone skull). In many places, the way the paper is written, the translation to human is described in a blurred manner, suggesting that it could be the case (there is even a reference to BMJ recommendations). This can not stay as it is as readers not familiar with the limitations of the technique might be mislead. All references or inputs for humans should be removed. If the Authors wish to make some translation, a specific section at the ned could be added with all the problems and issues for translation clearly exposed. 

Author Response

Thank you very much for this comment, we apologize for any confusing caused on our part. In order to clarify the confusing parts of the text, we have eliminated everything related to patients that could lead to misleading information in the animal part, and we have restructured some sections to make the article clearer. Furthermore, we have included a new specific section title “From preclinical models to clinical practice” summarizing the translation to clinical applications of ultrasound.

Reviewer 2 Report

In a current review article, the authors presented status of various ultrasound applications (diagnostic and therapeutic) in experimental animal models of stroke. They presented available animal data for ischemic and hemorrhagic stroke and the use of US.

I have following remarks:

  1. For contrast US please provide information on various contrast dyes available.
  2. If possible, give potential translation and clinical applications of animal data into human’s stroke. This would much strengthened current review paper.
  3. Consider a chapter on 3D/4D US.

Author Response

  1. For contrast US please provide information on various contrast dyes available.

Thank you for your observation, we have incorporated information on the characteristics and components of ultrasound contrast agents in the section  “2.1 Ultrasound monitoring in ischaemic stroke”.

  1. If possible, give potential translation and clinical applications of animal data into human’s stroke. This would much strengthened current review paper.

Thank you very much for this comment, we have added a new section title “From preclinical models to clinical practice” summarizing the translation to clinical applications of ultrasound.

  1. Consider a chapter on 3D/4D US.

Thank you very much for your suggestion, we have made subdivisions in section 5 tittle “Limitations and future of ultrasound”, adding a specific paragraph about 3D/4D in ultrasound. Now it appears as follows:

  1. Limitations and future of ultrasound

    5.1 Limitations of ultrasound

    5.2 Future of ultrasound

            5.2.1 New approaches of ultrasound

            5.2.2 Imaging 3D/4D in ultrasound